# Dual sequence definition increases the data storage capacity of sequence-defined macromolecules

Katharina S. Wetzel[1,3], Maximiliane Frölich[1,3], Susanne C. Solleder[1], Roman Nickisch[1], Philipp Treu[1] & Michael A. R. Meier [1,2✉]

Sequence-defined macromolecules offer applications in the field of data storage. Challenges include synthesising precise and pure sequences, reading stored information and increasing data storage capacity. Herein, the synthesis of dual sequence-defined oligomers and their application for data storage is demonstrated. While applying the well-established Passerini three-component reaction, the degree of definition of the prepared monodisperse macro-molecules is improved compared to previous reports by utilising nine specifically designed isocyanide monomers to introduce backbone definition. The monomers are combined with various aldehyde components to synthesise dual-sequence defined oligomers. Thus, the side chains and the backbones of these macromolecules can be varied independently, exhibiting increased molecular diversity and hence data storage capacity per repeat unit. In case of a dual sequence-defined pentamer, 33 bits are achieved in a single molecule. The oligomers are obtained in multigram scale and excellent purity. Sequential read-out by tandem ESI-MS/MS verifies the high data storage capacity of the prepared oligomers per repeat unit in comparison to other sequence defined macromolecules.

[1] Laboratory of Applied Chemistry, Institute of Organic Chemistry (IOC), Karlsruhe Institute of Technology (KIT), Straße am Forum 7, 76131 Karlsruhe, Germany. [2] Laboratory of Applied Chemistry, Institute of Biological and Chemical Systems—Functional Molecular Systems (IBCS-FMS), Karlsruhe Institute of Technology (KIT), Hermann-von-Helmholtz-Platz 1, 76344 Eggenstein-Leopoldshafen, Germany. [3] These authors contributed equally: Katharina S. Wetzel, Maximiliane Frölich. ✉email: m.a.r.meier@kit.edu

The rapidly growing field of sequence-defined macro-molecular synthesis has evolved to become one of the hot topics in polymer chemistry. Such macromolecules with a distinct order of the monomer units and a defined length are termed "uniform" molecules according to IUPAC[1]. Many routes towards sequence-defined structures have been developed over the last years, aiming for an increasing degree of control and precision[2–15]. The sequence-defined structures represent a new class of molecules in the polymer area, combining the features of classic polymers that are consisting of many repeating units, thus leading to their characteristic properties, with the ones of organic molecules, being uniform in size as well as constitution and topology. Thus, sequence-defined macromolecules pave the way for new applications ranging from enzyme mimicking, anti-counterfeiting tags or data storage[16]. The field is highly inspired by nature's precision synthesis, where exact structure enables functions like the storage and replication of the fundamental information of life in DNA[17–19]. However, nature's precision remains unreached by synthetic efforts to date.

Stepwise iterative approaches are a common route towards sequence-defined structures, since this step-by-step growth offers full control over the respective monomer units[4,5,9,13,20–31]. Multicomponent reactions are a well-established tool for achieving sequence-definition, as their modular character as well as their scalability and nearly quantitative yields, often without formation of any side products, are important features for the efficient synthesis of long sequences[3,22,23,32,33]. In particular, the Passerini three-component reaction (P-3CR) between a carboxylic acid, an aldehyde and an isocyanide, which was first reported in 1921 by the Italian chemist Mario Passerini[34], has been proven to be a powerful tool in the synthesis of sequence-defined macro-molecules of both high molecular weight and purity. For instance, it allowed for the synthesis of a sequence-defined decamer with nine different side chains[9] and variously functionalised macrocycles of different sizes[35].

Besides the investigation of enhanced synthetic approaches, which further increase the degree of control, the potential applications of such defined structures gain increasing attention[16]. Particularly, the application in the field of data storage has been discussed in detail[2,31,32,36–40]. The data storage capacity of DNA serves as prototype as it carries the genetic code[18,19]. In DNA, four nucleobases, arranged in a certain sequence order, define the "alphabet" that stores the fundamental information of each living organism. In contrast to nature, digital data storage in information technology is based on a binary code, consisting of "0"s and "1"s. Both systems have in common that data is stored by generating long sequences of the respective repeating units that form characteristic sequence orders, thus defining the code in which the information is stored. The number of permutations is a characteristic benchmark if different systems are to be compared. A sequence of eight binary digits corresponds to 1 byte = 8 bit = $2^8$ = 256 permutations. If the same number of permutations is stored in DNA, theoretically, only four nucleobases are required ($4^4$ = 256 permutations), however, the Shannon information capacity[41] is lower and one nucleotide corresponds to ~1.83 bits[42]. With regard to a potential application in data storage, the read-out of the encoded data is one of the crucial steps. To date, only a few approaches towards sequence-defined synthetic oligomers that allow for data storage[2,32] or encryption[43] have been developed. For the subsequent read-out, established analytic technologies, such as tandem mass spectrometry, have been reported to be convenient tools[16,38,44–46]. Regarding the data storage capacity, the number of independently selectable functional groups is of great importance. The first example using multiple functionalities within one repeating unit has recently been reported by our group[32]. Such an approach bears the potential of storing more information in shorter oligomers, i.e. obtained in less steps of the iterative synthesis cycle. Another interesting approach towards introducing multiple functionalities per repeating unit is the concept of dual side chain definition[23,47]. Recently, a strategy towards sequence-definition was introduced by the synthesis of sequence-defined oligothioetheramides with defined backbone, instead of the more common side chain definition[48]. Furthermore, an approach allowing for backbone or side chain variation was reported[49]. However, the variation of both backbone and side chain, at the same time, has not been demonstrated yet.

In the approach reported herein, the advantageous P-3CR is combined with a subsequent deprotection step. In combination with different specifically designed monomers, the approach allows variation of the backbone moiety, leading to backbone definition. Furthermore, we report an independent variation of both the backbone and the side-chain sequence, leading to dual-sequence-defined macromolecules offering a drastically enhanced structural versatility. Using this approach, a backbone-defined heptamer and a dual sequence-defined pentamer is synthesised, clearly demonstrating the potential of this new approach. The oligomers are obtained in high purity, as confirmed by SEC, mass spectrometry and NMR, while the synthesis is conducted in gram scale and allows for straightforward scale-up. Moreover, we report the read-out of the obtained backbone defined and dual sequence-defined macromolecules and discuss their suitability for applications in the field of data storage. The storage capacity of the newly synthesised oligomers is compared with the one of the previously reported sequence-defined decamer[9].

## Results and discussion

**Synthesis concept**. For the synthesis of dual sequence-defined oligomers, a previously developed approach taking advantage of the P-3CR was adapted[9]. Macromolecules were formed in a two-step iterative cycle consisting of the P-3CR and a subsequent deprotection step. Herein, not only the aldehyde component but also the applied monomer structure was varied in each iterative cycle, further increasing the structural variety. Thus, the aldehyde component defines different side chains and the additional variation of the monomer defines the backbone structure of these unique macromolecules. Therefore, dual sequence-defined macromolecules, in which the side chain and the backbone can be varied independently, are synthesised. The synthesis concept is outlined in Fig. 1. Using stearic acid **1** as starting acid component, linear macromolecules exhibiting different degrees of definition were synthesised by defining either the side chain[9], the backbone, or both the backbone and the side chain.

Side chain definition is achieved by variation of the aldehyde component, as demonstrated in earlier studies[9], demonstrating so far that nine different aldehydes are suitable components to introduce defined side chains. Combined with the herein utilised aldehydes, in total eleven different aldehydes **2a–k** were found to be suitable for side chain definition, including cyclic, branched and linear aliphatic moieties of different lengths, as well as aromatic compounds. For more details about the set of selectable aldehyde compounds and the structure of the previously reported decamer with defined side chains (side chain defined decamer = **S10**), please see the Supplementary Information (Supplementary Figs. 77 and 132). Backbone definition was achieved in a complementary manner by using different isocyanide monomers throughout the oligomer synthesis. Therefore, nine different monomers (**M1–M9**, see Fig. 2) were prepared and successfully applied in the synthesis of unique macromolecules featuring backbone definition. By variation of the isocyanide monomer, a backbone-defined heptamer was first

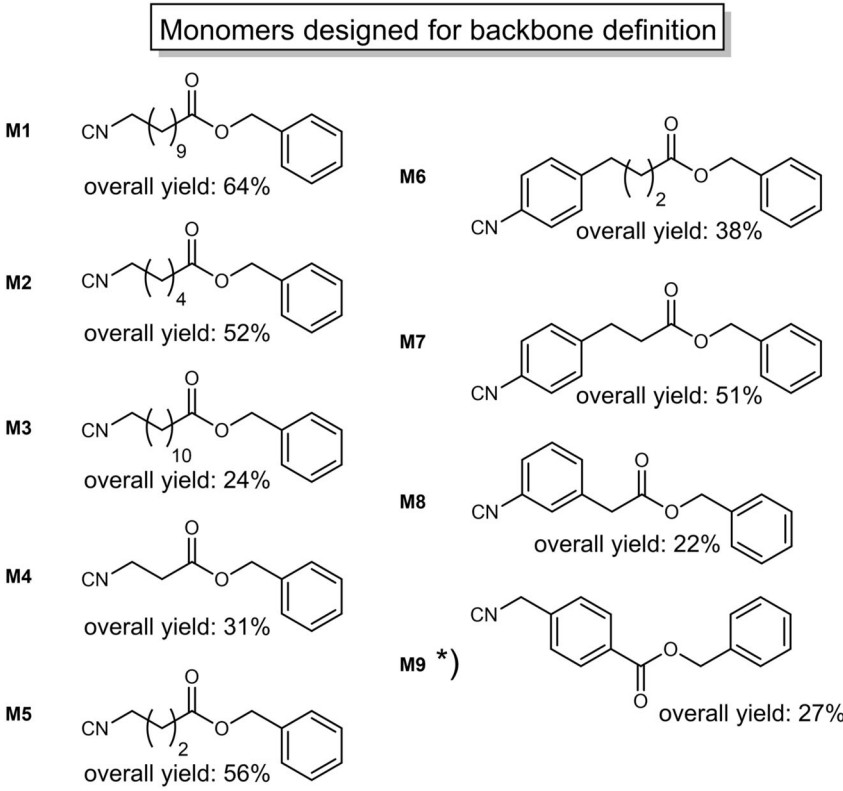

**Fig. 1 Synthesis strategy for dual sequence-defined oligomers.** The two-step iterative cycle allows for independent variation of side chain and backbone. The set of selectable side chains and backbone moieties is depicted in Supplementary Fig. 77.

**Fig. 2 Monomers M1–M9, prepared from the corresponding amino acids.** M1–M8 are prepared in three steps; *) in case of M9, the synthesis was performed via a four-step procedure. By variation of these monomers in the iterative synthesis cycle depicted in Fig. 1, different backbone moieties can be introduced to the macromolecules, thus leading to a backbone-defined sequence.

synthesised (backbone-defined monomer-heptamer = **B7**; backbone-defined oligomers of shorter length, i.e. intermediate product = **B1–6**). After this important proof of concept, the two types of sequence-definition were combined to obtain dual sequence-defined structures (**DS 1–5**). Thus, the monomer (**M1–M9**) as well as the aldehyde compound (**2a–k**) were varied independently in each iterative synthesis cycle.

**Monomer synthesis**. In total, a set of nine different monomers was established (see Fig. 2, **M1–M9**). All monomers were synthesised in three to four steps starting from the corresponding amino acids.

The synthesis was adapted from a procedure previously reported by our group (synthesis of monomer **M1**)[9]. Starting from different amino acids **3a–i**, the acid group was esterified with benzyl alcohol **4** using thionyl chloride **5** (Fig. 3). In a second reaction step, the previously formed ammonium salt **6** was N-formylated using trimethyl orthoformate **7**. The N-formamide **8** was then dehydrated using phosphoryl trichloride **9** forming the desired isocyanide benzyl ester monomer (**M1–8**). The synthesis of monomer **M2** is shown exemplarily in Fig. 3.

The monomers were obtained in moderate to high overall yields of up to 64%, except for **M9** (see discussion below), and in excellent purity. All the different monomers were thoroughly

**Fig. 3 Three-step synthesis procedure of monomer M2.** The synthesis starts with the esterification of the amino acid, followed by the N-formylation and a final dehydration step. The procedure was applied in the synthesis of monomers M1–M8.

characterised by proton and carbon NMR, mass spectrometry, and IR spectroscopy in order to verify their structure and purity (see Supplementary Figs. 1–77 and Supplementary Table 1). Before utilising the monomers in the oligomer synthesis, they were also analysed by SEC, which is equipped with columns specifically designed for measuring low molecular weight compounds. The syntheses were performed on multigram scale of up to 15 grams. Applying the three-step synthesis protocol, seven new monomers (**M2–M8**, **M1** was previously reported)[9] were successfully synthesised. If the acid functionality is directly connected to an aromatic ring, as it is the case for 4-(aminomethyl)benzoic acid **3i**, the starting material for **M9**, the esterification reaction with benzyl alcohol **4** did not proceed as expected. In order to circumvent the low reactivity that led to unsatisfactory yields, a fourth reaction step had to be introduced (see Supplementary Methods for detailed information). Thus, in the first step, an esterification was performed using methanol **10** as solvent instead of the benzyl alcohol **4** to yield the methyl ester **11**. After the N-formylation step, the additional reaction step, being a transesterification with benzyl alcohol **4**, was carried out before dehydration to the targeted isocyanide **M9**. The overall yield in the synthesis of **M9** is thus lower compared to the other monomers.

The results of the synthesis of the monomers **M1–M9** as well as all intermediates are summarised in Supplementary Table 1, an overview of the available monomers and aldehyde components is provided in Supplementary Fig. 77. Having the monomers **M1–M9** in hands, the reactivity of the monomers was investigated in a P–3CR with a subsequent deprotection in order to ensure their potential usefulness in the oligomer synthesis, as depicted in Fig. 1.

**Oligomer synthesis**. Since it has been demonstrated before that long sequences with or without side chain variation can be achieved in high overall yields and multigram scale by applying monomer **M1**[9,35], the next step towards dual sequence-definition was to investigate if the backbone can be varied by utilising **M1–M9**. Thus, first an oligomer with a sequence-defined backbone (**B1–7**) was synthesised by varying the monomer in each reaction cycle (see Supplementary Methods for detailed experimental procedures). The aldehyde component, on the other hand, was not varied (isobutyraldehyde **2a** was used throughout) for the synthesis of **B1–7**. The reaction conditions were adapted from previous protocols, utilising the P–3CR with subsequent deprotection[9,35]. Stearic acid **1** was used as starting material. The aldehyde **2a** and the monomer components **M1–9** were used in

small excess relative to the carboxylic acid. The reactions were stirred at room temperature for 24 h and the products were obtained after purification by column chromatography. According to previous observations, which showed a slowing down of the reaction as the oligomer grew larger[3], the reaction time was increased to 48 h for longer sequences (see Supplementary Methods). The first P-3CR product **B1** was obtained in excellent yield (98%) and purity, as evidenced by proton and carbon NMR, mass spectrometry, IR spectroscopy and SEC. The following deprotection step was performed in ethyl acetate using palladium on activated charcoal as heterogeneous catalyst and purging the mixture with hydrogen overnight. Full conversion was ensured by TLC monitoring and SEC analysis, which showed a shift towards higher retention times, thus lower molecular weight, as expected. Furthermore, the cleavage of the benzyl ester was observed in the [1]H NMR spectrum. This reaction sequence was repeated, using a different monomer in each iterative cycle. From the fifth deprotection on, the solvent was changed from ethyl acetate to THF, due to solubility issues. The deprotected acid of the growing oligomers was barely soluble in ethyl acetate, thus hindering the successful deprotection, as the reaction mixture became too viscous. As both the protected and the deprotected oligomers were soluble in THF, it proved to be a suitable solvent for the reaction. After each reaction step, full characterisation was performed using NMR, SEC, mass spectrometry and IR spectroscopy (See Supplementary Figs. 78–107, Supplementary Table 2). By iteration of the reaction cycle, defined sequences up to a backbone-defined heptamer **B7** were obtained. The structure as well as SEC and ESI-MS characterisation are shown in Fig. 4, clearly confirming the successful synthesis. All additional characterisation data can be found in the Supplementary Information (Supplementary Figs. 78–107 and Supplementary Table 2). The overall yield of the heptamer **B7** was 31% over 14 reaction steps, which is lower compared to the previous monomer approach, where the side chain was varied (44% for a decamer with side chain variation **S10** after 19 reaction steps)[9].

Reasons for the lower yields are probably the aromatic scaffolds of the monomers with different intrinsic reactivity (see also monomer synthesis discussion above) and the increasing polarity with growing length, rendering the purification by column chromatography increasingly difficult. However, the backbone-defined heptamer **B7** clearly demonstrated the suitability of the different monomers for achieving backbone definition, thus increasing the degree of structural variability as well as the general versatility of the P–3CR monomer approach.

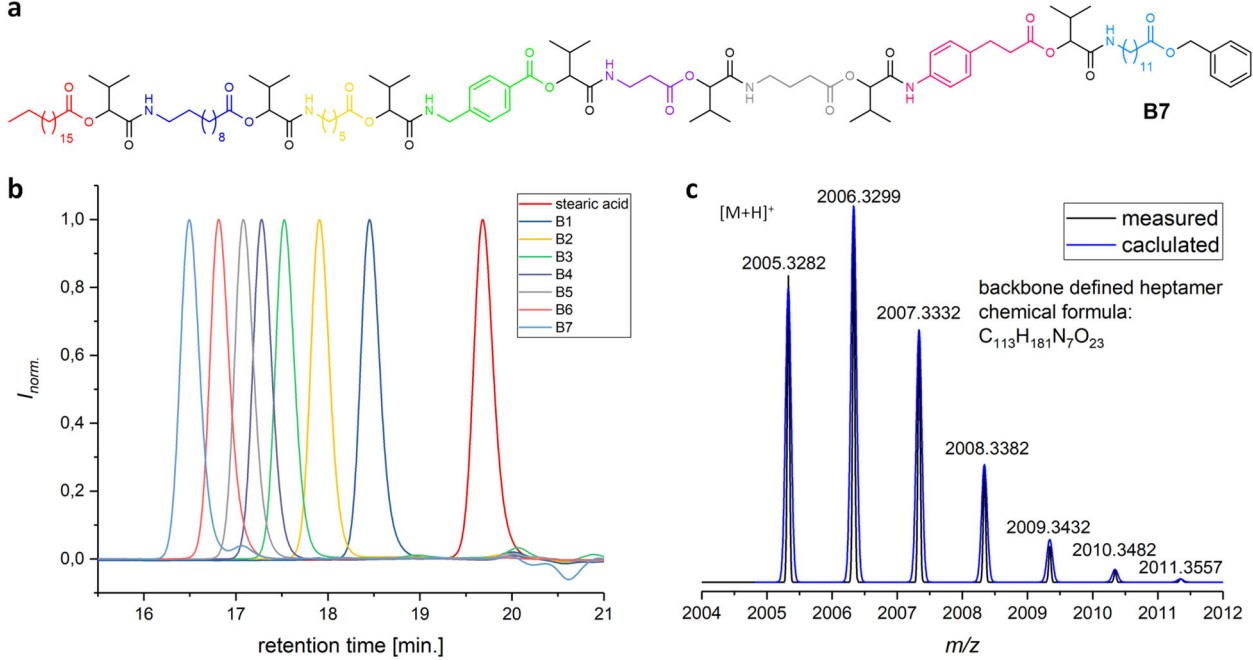

**Fig. 4 Characterisation of the backbone-defined heptamer B7. a** Chemical structure of **B7**. **b** SEC results after each P-3CR reaction of the iterative synthesis cycle, verifying the high purity of the products. The colour code identifies the respectively used backbone moieties (Fig. 4a). **c** High-resolution ESI-MS measurement: calculated and observed isotopic pattern of singly protonated product **B7**. See Supplementary Methods for further characterisation of the products after each reaction.

Consequently, the two types of definition achieved by this approach, e.g. side chain and backbone definition, were next combined in order to introduce different defined side chains as well as backbone moieties into each repeating unit. Therefore, a new oligomer series (**DS1–5**) was synthesised, while varying both the AB-monomer and the aldehyde component independently in each step (see Supplementary Methods for detailed experimental procedures). Again, stearic acid **1**, **M1** and isobutyraldehyde **2a** were used as starting materials in the first P–3CR to obtain **DS1**, similarly as described for **B1**. In the deprotection step, THF was used as solvent from the beginning of the oligomer synthesis to ensure solubility of the growing oligomer.

In Fig. 5, the structure of the obtained dual sequence-defined pentamer (**DS5**) and the SEC results of all intermediate reaction steps are shown. Besides the characterisation by SEC, the oligomers were also analysed by proton and carbon NMR, by IR spectroscopy, and mass spectrometry after each reaction step (see Supplementary Figs. 108–128 and Supplementary Table 3 for full characterisation). The SEC traces clearly prove the purity of the oligomers. From the pentamer stage on, a small shoulder towards higher retention time, thus lower molecular weight, was observed, which could not be separated. The purity of **DS5** was 91%. It is noteworthy that also in the backbone-defined oligomer, the shoulder arose when the aliphatic monomer with the longest carbon chain (**M3**) was utilised (compare Supplementary Figs. 106 and 128). Apparently, the problems are caused by minor undetectable impurities in the monomer compound **M3** and thus for future syntheses, the use of this specific monomer should be avoided.

**Sequential read-out by tandem ESI-MS/MS.** A possible application of the sequence-defined macromolecules is in the field of data storage, and thus a sequential read-out by tandem ESI-MS/MS was investigated. The obtained results were subsequently thoroughly compared with each other with regard to data storage capacity (see next paragraph).

Tandem ESI-MS/MS fragmentations of all the synthesised oligomers were performed to identify the predominant fragmentation patterns. Interestingly, it was found that for all the oligomers, independent of the degree of definition, the same predominant fragments were found (next to the carbonyl and next to the ester group, see Supplementary Fig. 129 and Supplementary Note 1). This was an important result, as it verified that even in case of dual sequence-definition, the fragments do not get too complex for manual analysis and thus read-out of the sequence remains feasible. In Fig. 6, the fragmentation next to the carbonyl group from both ends of the dual sequence-defined pentamer **DS5** is depicted (i.e. the sequence can be read from the starting stearic acid moiety or from the benzyl protected last monomer introduced). The information is complementary and provides an error-proof mechanism.

As depicted in Fig. 6, the fragmentation follows distinct rules (see above, next to the carbonyl and next to the ester group and Supplementary Fig. 129) and by recombining the masses of the different fragments, the initial structure can be re-established. Therefore, the start and the end fragments were first identified. By adding the masses of the possible middle fragments, which are a combination of the Passerini moiety with the respective backbone moieties and side chains, and adding the result to respective masses of the start and end fragments, the mass of the molecule can be calculated and used to identify the intact molecule in the MS spectrum. Similarly, the masses of the fragments can be calculated by the general formula provided in the Supplementary Information (Sequencing rules, see Supplementary Eqs. 1–6). Thus, the masses of the initial structure as well as the expected fragments can be calculated and traced in the MS/MS spectra, enabling the successful read-out of the sequence. It is important to note that both fragmentation pathways as well as the formulas provided in the Supplementary Information are generally applicable to all herein discussed sequence-defined macromolecules, thus underlining the robustness and generality

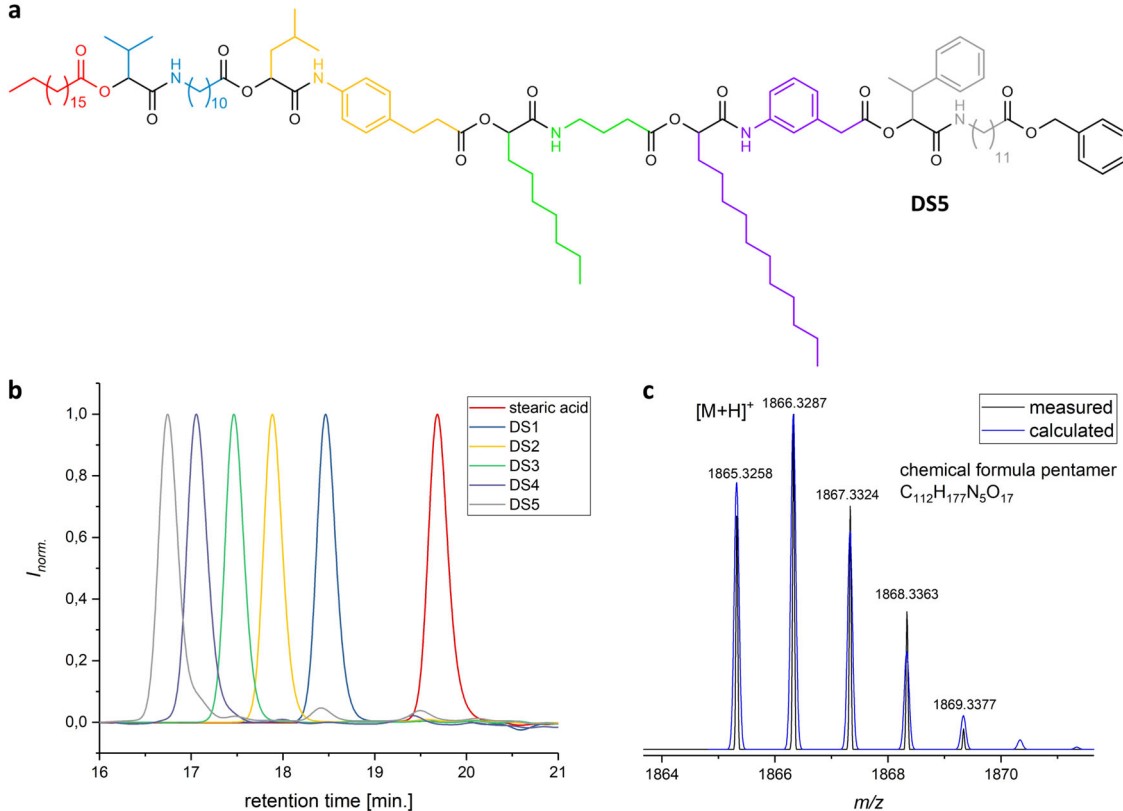

**Fig. 5 Characterisation of the dual sequence-defined pentamer DS5. a** Chemical structure of **DS5**. **b** SEC results after each P-3CR reaction of the iterative synthesis cycle, verifying the high purity of the products. The colour code identifies the respectively used side-chains and backbone moieties (Fig. 5A). **c** High-resolution ESI-MS measurement: calculated and observed isotopic pattern of singly protonated product **DS5**. The corresponding sodium ion was also found (see Supplementary Methods, also for further characterisation of the products after each reaction).

of the read-out. One calculation example applying the herein established sequencing rules is provided in the Supplementary Methods for **DS5** (see Supplementary Fig. 130).

Besides the fragmentation next to the carbonyl, another fragmentation pattern next to the ester group was also observed, providing a further possibility to check for reading errors. Furthermore, the middle parts of the fragments (without starting and end moieties) of both fragmentation types were observed. Other prominent fragments, that are, for the sake of clarity, not assigned in Fig. 6 belong to these middle fragments and to the fragmentation next to the ester and can be assigned analogously. A detailed analysis of the middle fragments as well as of the second fragmentation type (for possible error-proofing) are provided in the Supplementary Information (Supplementary Figs. 135–137). All above mentioned oligomers (**S5**, **B5**, **DS5** and **B7**) and the side chain defined decamer **S10** from our previously reported work[9] were thoroughly sequenced in the same manner, details are provided in the Supplementary Information (see Supplementary Figs. 131–137).

**Comparison of data storage capacity of different systems and discussion**. Having established the writing as well as reading of our macromolecules, it is important to compare their individual maximal data storage capacity to one another. As discussed in the introduction, information technology is based on the binary code of "0"s and "1"s, originating from switches and triggers "off" and "on". To store information in such a system, very long sequences are required, since the binary system exhibits a small data storage capacity of one bit per repeating unit. In the binary system, a sequence of eight binary digits represents 1 byte, which is 8 bit or $2^8$, thus 256 permutations. In a quaternary system, on the other

hand, shorter sequences are required to achieve the same number of permutations: $4^4 = 256$, thus, only tetramers would be required to achieve the same storage capacity. In Fig. 7, the number of achievable permutations of five different tetramers (digital binary code, quaternary DNA code and the herein presented side chain, backbone and dual sequence-definition), selected as examples for demonstration reasons, are compared to each other.

Due to their modular character, multicomponent reactions are convenient tools for applications in the field of data storage as the different components can introduce a large variety of different moieties into a growing oligomer and thus naturally increase structural variety and finally the data storage capacity. In this context, it is important to note that two essential criteria have to be fulfilled regarding the selectable moieties (i.e. side chain and backbone) for successful synthesis (i.e. data storage) and subsequent read-out: first, they have to exhibit orthogonal and sufficient reactivity in the oligomer synthesis. Second, they have to show distinct fragmentation patterns und unique mass-fragments in tandem ESI-MS/MS experiments that are distinguishable from all other fragments enabling unambiguous assignment and reestablishment of the initial structures. Of course, the herein used components and oligomers were selected considering these aforementioned criteria and are thus suitable for data storage, as demonstrated above.

The synthesis of a side chain defined decamer **S10**, allowing the introduction of 10 different side chains, was reported before by our group (please note that the sequence-defined decamer carries ten side chains in total, but nine different ones. Thus, a set of nine side chains was established)[9]. Together with the previously reported approaches[9,35], a set of 11 possible side chains was thus established in total herein. If this set of commercially available

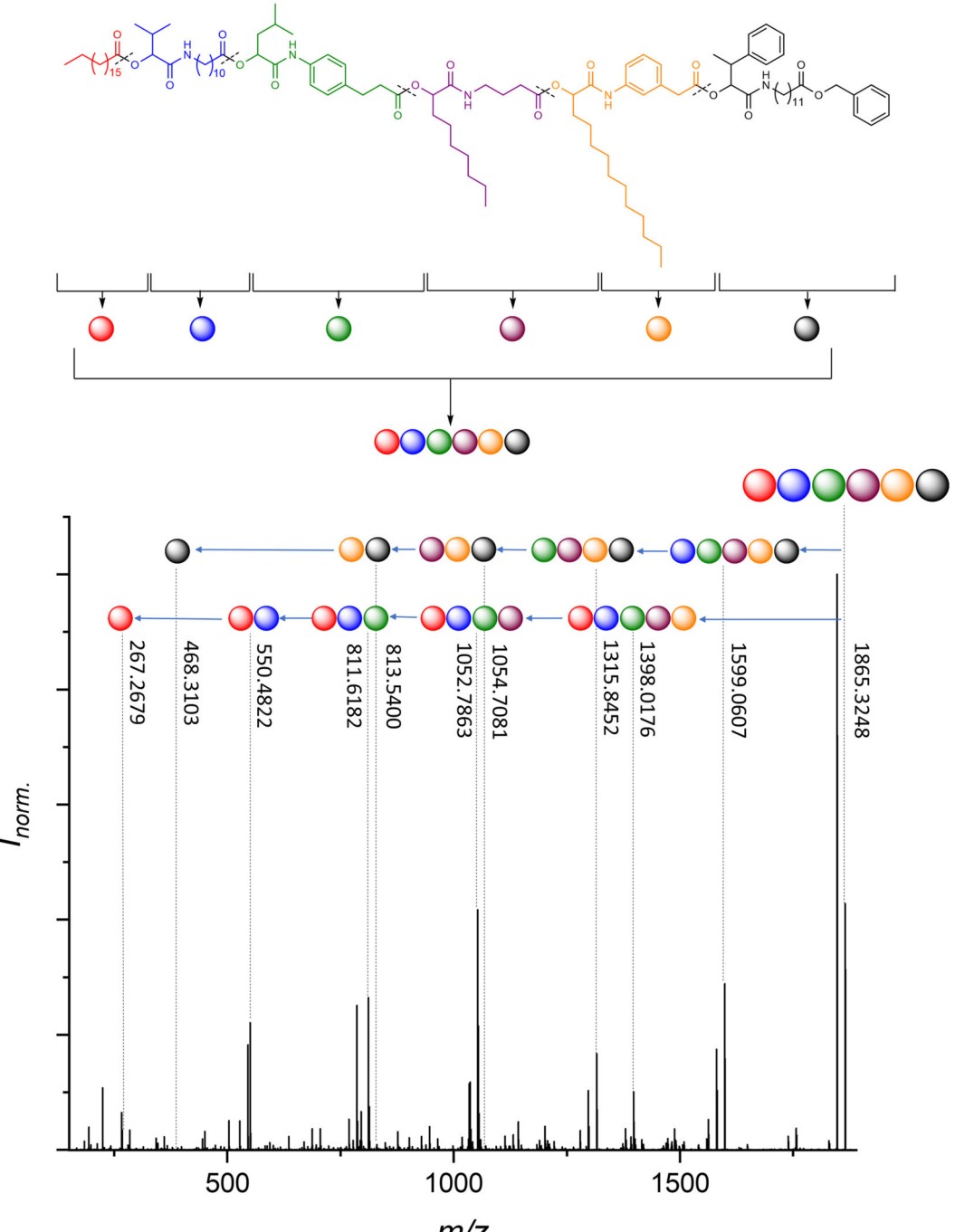

**Fig. 6 Sequential read-out of the dual sequence-defined pentamer DS5.** Read-out is achieved via fragmentation by tandem mass spectrometry, revealing the expected fragmentation pattern. In the spectrum, the fragmentation next to the carbonyl group from both ends of the oligomer is depicted. By recombining the fragments, the initial structure of the pentamer can be re-established and thus the stored information is read. Assignment of the middle fragments as well as the second fragmentation pattern are provided in Supplementary Figs. 135–137.

aldehyde components **2a–k** (see Supplementary Fig. 77) was applied for the synthesis of a tetramer with 11 possible and varying side-chains, this would correspond to $11^4 = 14.641$ permutations (i.e. 13.84 bits, see Fig. 7), representing an enormous increase in the storage capacity if compared to DNA. In case of only backbone variation, a slightly smaller number of permutations is achieved since the library of available monomers is smaller (i.e. **M1–M9**). Thus, a backbone-defined tetramer would represent $9^4 = 6.561$ permutations or 12.68 bits (see Fig. 7).

However, the number of permutations can significantly be increased by combining the two types of sequence-definition. Thus, by applying the dual sequence-defined macromolecules for data storage, this number can be significantly extended, since the molecular diversity is significantly increased. Eleven possible side chains plus nine possible backbone moieties lead to $(11 \times 9)^4 = 96.059.601$ permutations (26.52 bits, see Fig. 7) in case of a tetramer. This is a highly notable increase compared to the previous systems described in the literature.

The length of the sequence has an even larger influence on the data storage capacity than the molecular diversity, as Eq. 1 suggests.

$$\left( n_{permutations} \right) = \left( n_{variations\ per\ repeat\ unit} \right)^{degree\ of\ oligomerisation} \qquad (1)$$

Dual-sequence-defined pentamers, such as the above discussed **DS5**, thus represent $(11 \times 9)^5 = 9.509.900.499$ permutations (33.15 bits, 4.14 bytes). The permutations can be translated into bit and byte terminology by the following equations:

$$bit = \frac{\log \left( n_{permutations} \right)}{\log(2)} \qquad (2)$$

and

$$8\ bits = 1\ byte \qquad (3)$$

Thus, **DS5** represents 4.14 bytes storage capacity. The above discussion is of theoretical nature and the discussed storage capacities can only be claimed if a read-out is unambiguously demonstrated, as already demonstrated above, see also Supplementary Information.

The data storage capacity of the herein presented oligomers is furthermore thoroughly compared in Table 1, which impressively illustrates the influence of the sequence length and the number of selectable functionalities per repeating unit on the number of permutations and thus their storage capacity in bits. The storage capacity can either be increased by elongation of the sequence or by increasing the degree of definition. As Eq. 1 suggests, the length of the oligomer has a larger influence on the storage capacity, but this is synthetically often more difficult to achieve. Thus, as the synthesis of very long and chemically pure sequences (DP over 15) in larger amounts gets more and more challenging, an increase in the number of possible variations in the monomer structure can help circumvent this issue, at least to a certain extent. Dual sequence-definition thus offers a significant improvement in data storage capacity, while minimising the synthetic effort drastically. The data storage capacity of the dual sequence-defined pentamer **DS5** is in the same range as the one of the side chain defined decamer **S10** (see Table 1), while the synthetic workload is reduced by half for the oligomer synthesis.

In the foreseeable future, the data storage capacity will be further increased by increasing the number of potential backbone and side chain moieties. If, for example, a library of 20 monomers as well as 20 commercially available aldehyde components would be used for the synthesis of a dual sequence-defined decamer, the number of permutations would increase by factor of $1.78 \times 10^{16}$ to $(20 \times 20)^{10} = 1.05 \times 10^{26}$ permutations, which would translate to 86.4 bits and thus to 10.8 bytes. The data storage capacity would thus increase by factor 2.66. This realistic example unambiguously visualises the current and future potential of dual sequence-defined macromolecules for applications in the field of data storage. The choice of both monomers and side-chains will have to fulfil two important criteria: (i) suitable orthogonal and sufficient reactivity in the iterative growth cycle and (ii) distinguishable and distinct fragmentation patterns in MS/MS experiments. Of course, our herein used components and monomers were selected and designed using these two basic rules. Moreover, the read-out of longer sequences in combination with larger libraries of selectable side chains and backbones will also require the design and implementation of sophisticated automated data analysis tools, as analysis by hand, as described herein, is laborious and will not be time-efficient for longer or more complicated sequences.

In summary, we demonstrated that a set of nine structurally diverse monomers can be applied in the synthesis of sequence-defined macromolecules by first establishing a procedure for backbone-definition using the P-3CR reaction. Furthermore, we herein report the synthesis of dual sequence-defined macromolecules, having independently selectable side chains as well as backbone moieties. The products were obtained in high purity, as evidenced by various characterisation techniques, and the synthesis was conducted in multigram scale. Thus, structural diversity in sequence-defined materials was significantly enhanced. The dual sequence-defined oligomers were successfully applied for sequential read-out via tandem mass spectrometry and compared to other sequence-defined oligomers, significantly increasing the data storage capacity compared to only side-chain or backbone-defined oligomers. Thus, a data storage capacity of more than 33.15 bits in one pentamer was demonstrated using this approach. Two characteristic fragmentation patterns were identified, simplifying the analysis of complex and long sequences, and also providing possibilities for error correction during the read-out.

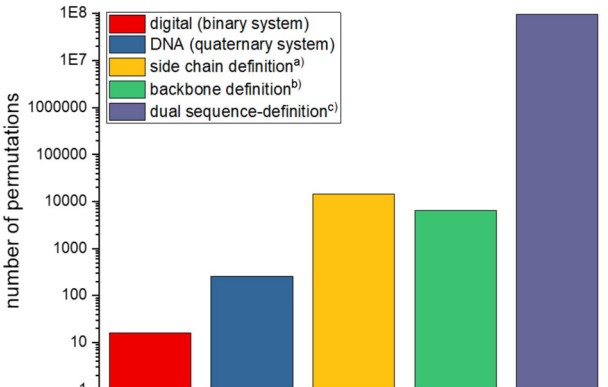

**Fig. 7 Comparison of different data storage systems.** Artificial (i.e. digital, binary) and naturally (i.e. DNA, quaternary) applied data storage systems are compared with the herein discussed macromolecular data storage systems. The number of permutations of five tetramers are depicted. Calculations are based on [a]11 possible side chains, demonstrated herein; [b]9 monomers introduced herein; [c]the combination of these 11 possible side chains and 9 possible monomers. Please note the logarithmic scale.

**Table 1 Comparison of the herein demonstrated data storage capacity.**

|  | Pentamer (S5)[a] | Pentamer (B5)[b] | Pentamer (DS5)[c] | Decamer (S10)[a] | Heptamer (B7)[b] |
|---|---|---|---|---|---|
| Permutations | 161.051 | 59.049 | 9.509.900.499 | 25.937.424.601 | 4.782.969 |
| Bit | 17.30 | 15.85 | 33.15 | 34.59 | 22.19 |
| Byte | 2.16 | 1.98 | 4.14 | 4.32 | 2.77 |

Three pentamers (exhibiting different degrees of definition) and two longer oligomers are compared, illustrating the influence of the number of selectable functionalities per repeating unit and of the sequence length on the achievable permutations, bits and bytes.
[a]Calculated using the established 11 possible side-chains demonstrated so far.
[b]Calculated based on the 9 monomers, i.e. **M1–M9**, demonstrated herein.
[c]Calculated based on 11 possible side chains and 9 possible monomers.

## Methods

**Monomer synthesis procedure for monomers M1—M8.** Detailed synthesis procedure for **M9**, as well as characterisation and detailed synthetic procedures of all monomers are provided in the Supplementary Information.

Esterification: The respective amino acid (1.00 eq.) was suspended in THF, benzyl alcohol (12.0 eq.) was added and the mixture was cooled to 0 °C. Thionyl chloride (3.10 eq.) was added dropwise at 0 °C. After the addition, the reaction mixture was warmed to room temperature and stirred overnight. The solution was poured into 500 mL of diethyl ether and stored in the freezer for 1 h. The product was filtered off and another 500 mL of diethyl ether were added and the suspension was stored in the freezer for another hour. The product was filtered off and dried under high vacuum.

N-formylation: Trimethyl orthoformate (10.0 eq.) was added to the benzyl ester ammonium chloride (1.00 eq.) and the mixture was heated to 100 °C to dissolve the solid. The reaction was heated under reflux overnight, then cooled to room temperature and the trimethyl orthoformate was removed under reduced pressure to afford the crude product which was used without further purification.

Dehydration: The N-formamide (1.00 eq.) was dissolved in DCM (0.3 M). Diisopropylamine (3.00 eq.) was added and the reaction mixture was cooled to 0 °C. Subsequently, phosphoryl trichloride (1.20 eq.) was added dropwise and after the addition, the mixture was stirred for 2 h at room temperature. For quenching the reaction, a solution of sodium carbonate (20 wt%, 75 mL) was added at 0 °C. After stirring the heterogeneous mixture for 30 min at room temperature, 50 mL of DCM and 50 mL of water were added. The aqueous phase was separated and the organic layer was washed with water (3 × 80 mL) and brine (80 mL). The organic layer was dried over sodium sulfate and the solvent was removed under reduced pressure. The product was purified via column chromatography (cyclohexane/ethyl acetate).

**Typical two-step iterative cycle for oligomer synthesis.** Detailed synthesis procedures for each step of the oligomer syntheses as well as full characterisation are provided in the Supplementary Information.

*Passerini reaction.* The acid (1.00 eq.) was dissolved in DCM (1.00 M–0.300 M). Subsequently, the aldehyde component (1.50 eq.–3.00 eq.) and the monomer (1.50 eq.–3.00 eq.) were added and the reaction mixture was stirred at room temperature for 24 to 48 h. The solvent was removed under reduced pressure and the crude product was purified by column chromatography (cyclohexane/ethyl acetate). The excess of the monomer can be recovered during purification.

*Deprotection.* The benzyl ester (1.00 eq.) was dissolved in THF (0.500 M–0.200 M). Subsequently, palladium on activated charcoal (10 wt%) was added and the mixture was purged with hydrogen (2–4 balloons) and stirred under hydrogen atmosphere overnight. The heterogeneous catalyst was filtered off and the solvent was evaporated under reduced pressure to obtain the desired product.

For instruments, materials, conduction and evaluation of tandem ESI-MS/MS experiments, full characterisation of all synthesised substances and more detailed synthesis procedures, please see Supplementary Information.

**Reporting summary.** Further information on research design is available in the Nature Research Reporting Summary linked to this article.

## Data availability

All relevant data is included as supplementary information and is also available from the corresponding author.

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

## Acknowledgements

K.S.W., M.F., R.N. and M.A.R.M. would like to acknowledge funding by the German Research Council (DFG) in the context of the SFB 1176 (project A3). We gratefully acknowledge Katja Krell and Yixuan Jia for synthetic support and the analytical team for analytical support. Prof. Barner-Kowollik (KIT) and his group are acknowledged for access to SEC-ESI-MS equipment.

## Author contributions

All authors contributed to the discussion. K.S.W., M.F. and M.A.R.M. conceived and designed the project. K.S.W., M.F. and S.C.S. designed the experiments with input from M.A.R.M. K.S.W., S.C.S., R.N. and P.T. synthesised the monomers. K.S.W. synthesised the oligomers. M.F. performed the ESI-MS/MS experiments. K.S.W., M.F. and M.A.R.M. analysed and interpreted the data. K.S.W. and M.F. prepared the figures. K.S.W. wrote the paper with feedback from all the authors.

## Competing interests

The authors declare no competing interests.
