## [Peer Review File · Communications Chemistry]

This manuscript has been previously reviewed at another Nature Research journal. This document only contains reviewer comments and rebuttal letters for versions considered at Communications Chemistry.

Reviewers' comments:

Reviewer #1 (Remarks to the Author):

This manuscript was first submitted to Nature Communications and is now considered for publication in Communications Chemistry, so the present report also relies on comments from the first reviewing round.

Despite writing in their rebuttal that reviewers' comments certainly helped to further improve the quality of their manuscript, it should first be acknowledged that the authors performed only very minor changes to produce the new version of the main text. Some previously raised issues still need to be addressed, as well as a few new remarks.

The whole discussion on storage capacity would deserve a dedicated section, instead of "interfering" with MS/MS. The Shannon's information theory is mentioned on page 3 to explain that storage capacity of nucleotides is lower than expected from calculation. I am really unfamiliar with this mathematical approach so my following question might be very naive: would the Shannon's theory also apply to the oligomers described in this manuscript and, if so, what would be their actual (corrected) storage capacity?

As mentioned by the authors in their responses to comments "Data storage can only be claimed if both writing and reading are demonstrated". I fully agree with the statement if "reading" is replaced by "blind reading". Retrieving a code from MS/MS data when the code is known is not so complicated, but what about reading an unknown? The authors should demonstrate sequence reconstruction of a pretended unknown oligomer by explaining rules to be used for MS/MS data analysis, that is, detailing mass shifts to be considered to search for the next fragment as a function of the combined backbone and side-chain code in repeating units.

It is true that MS/MS sequencing can be properly performed without detailed knowledge of dissociation mechanisms leading to the detected fragments. This is named "sequencing rules" that mainly describe how mass calculation should be performed to search for successive ions in one fragment series to be able to safely reconstruct the sequence from one or the other chain end. As mentioned in the previous comment, these rules should be described so that the reader can evaluate their robustness. In contrast, without knowing dissociation mechanisms at work, one cannot propose reliable structures for fragments. Accordingly, structures proposed in section 3.5 of SI are wrong and should be removed. Instead, fragment composition can be described as a combination of different repeating units plus additional atoms that presumably compose their end-groups, so that accurate mass data can be used to support ion assignments.

MS/MS spectra are experimental data, not schemes. So, as previously recommended by one reviewer, MS/MS peaks used for sequence reconstruction should be labeled with m/z values. In their response to comments, the authors argue that adding m/z values in MS/MS spectra would make the figures confusing and, instead, list these values in separate tables. This is not a convenient presentation to the reader. Moreover, in the only MS/MS spectrum reported in the manuscript (Figure 5), there are only 11 useful peaks to be labeled with their respective m/z values, so adding these values will not complicate the figure that much (no need for 4 digit values, one decimal will be sufficient since accurate mass measurements are reported elsewhere).

Although most errors have been appropriately corrected, there are still a few issues with accurate mass data reported in SI (as listed below) and, although they are related to monomer synthesis, they must be fixed. In contrast, how can the authors explain that correcting calculated values had an influence on measured values? For example, changing the calculated m/z value of the 6d ion (p S20) from 180.1103 to 180.1019 has also changed the experimental 180.1104 to 180.1020!!! The same remark applies for M4 on p S22 and for 6f on p S32.

- p S23: elemental composition of this ion is wrong: this should be $C_{11}H_{16}NO_2^+$, not

C11H15NO2+

- p S25: the elemental composition is reported with no charge: do they author mean that this ion is not a protonated molecule but a molecular ion? If so, the charge should be "dot plus".
- p S37: the reported experimental m/z value lacks precision.
- p S39: calculated value for C33H56NO5 is 546.4153, not 546.4158.
- p S55: same remark as on p S25.

Reviewer #1

Comments: This manuscript was first submitted to Nature Communications and is now considered for publication in Communications Chemistry, so the present report also relies on comments from the first reviewing round.

Despite writing in their rebuttal that reviewers' comments certainly helped to further improve the quality of their manuscript, it should first be acknowledged that the authors performed only very minor changes to produce the new version of the main text. Some previously raised issues still need to be addressed, as well as a few new remarks.

Answer: We thank the reviewer for the assessment and have taken the constructive comments into consideration. We have addressed the concerns and suggestions point-by-point below. The comments certainly helped to further improve the quality of our manuscript. Changes to the manuscript or within the supporting information that result from the first revision are marked in yellow. Changes with this second revision are marked in green.

Comments: The whole discussion on storage capacity would deserve a dedicated section, instead of "interfering" with MS/MS.

Answer: Thank you for this remark. We have now restructured the manuscript accordingly and show two separate sections for MS/MS and data storage capacity. We start with the MS/MS sequencing and discuss data storage afterwards in the new section "Comparison of data storage capacity of different systems and discussion." Please note that this was achieved by reorganizing already existing parts of the manuscript and thus not many "green" parts are visible.

Comments: The Shannon's information theory is mentioned on page 3 to explain that storage capacity of nucleotides is lower than expected from calculation. I am really unfamiliar with this mathematical approach so my following question might be very naive: would the Shannon's theory also apply to the oligomers described in this manuscript and, if so, what would be their actual (corrected) storage capacity?

Answer: The Shannon information capacity describes the amount of information that can be transmitted faultlessly by a given communication channel. It describes the highest possible information rate (or bit rate) that can be transmitted without any mistakes. The information content of a message is a logarithmic size which indicates how much information can be transmitted in a message. In our work, the synthesised oligomers are the information channel, thus, information is read by sequencing them and by decoding the sequenced data. In case of DNA, the Shannon information capacity has been discussed several times and various errors, such as experimental factors and sequencing errors, just to name a few of them, have been taken into account to calculate the Shannon information capacity. The mentioned errors make the communication channel noisy and thus lower the actual information content. In theory, this can be calculated for any kind of communication channel. However, the parameters being applied for DNA do not apply for our system. Therefore, it is not possible at this stage to simply transfer the concept and apply it to our study. Instead it would be necessary to identify possible errors and to define the impact of each error, which is out of scope for our study and will most likely become a problem when reading longer or structurally even more complicated

sequences. Furthermore, we would like to point out that in our work, the maximum data storage capacity is discussed, and that one needs to differentiate between the two terms data storage capacity and information content.

Comments: As mentioned by the authors in their responses to comments “Data storage can only be claimed if both writing and reading are demonstrated”. I fully agree with the statement if “reading” is replaced by “blind reading”. Retrieving a code from MS/MS data when the code is known is not so complicated, but what about reading an unknown? The authors should demonstrate sequence reconstruction of a pretended unknown oligomer by explaining rules to be used for MS/MS data analysis, that is, detailing mass shifts to be considered to search for the next fragment as a function of the combined backbone and side-chain code in repeating units.

Answer: We have added one section to the manuscript, explaining how read-out is performed in a highly detailed fashion, as well as one section to the supporting information, demonstrating the sequence reconstruction in detail. We do agree that is important information to the reader. With the described information “blind reading” is unambiguously possible. To achieve this, it was important to identify characteristic fragmentation patterns. Without having identified such characteristic fragmentation patterns, it would never be possible to do any read-out.

Comments: It is true that MS/MS sequencing can be properly performed without detailed knowledge of dissociation mechanisms leading to the detected fragments. This is named “sequencing rules” that mainly describe how mass calculation should be performed to search for successive ions in one fragment series to be able to safely reconstruct the sequence from one or the other chain end. As mentioned in the previous comment, these rules should be described so that the reader can evaluate their robustness. In contrast, without knowing dissociation mechanisms at work, one cannot propose reliable structures for fragments. Accordingly, structures proposed in section 3.5 of SI are wrong and should be removed. Instead, fragment composition can be described as a combination of different repeating units plus additional atoms that presumably compose their end-groups, so that accurate mass data can be used to support ion assignments.

Answer: Thank you for this comment. We have removed the proposed structures in the supporting information and have added a section describing the sequencing rules in detail. We have added examples of the sequence reconstruction. Moreover, in the main manuscript, we have added a paragraph describing the sequence reconstruction:

In Figure 4, the fragmentation next to the carbonyl group from both ends of the dual sequence-defined pentamer **DS5** is depicted (i.e. the sequence can be read from the starting stearic acid moiety or from the benzyl protected last monomer introduced). The information is complementary and provides an error-proof mechanism. As depicted in Figure 4, the fragmentation follows distinct rules (see above, next to the carbonyl and next to the ester group) and by recombining the masses of the different fragments, the initial structure can be re-established. Therefore, the start and the end fragments were first identified. By adding the masses of the possible middle fragments, which are a combination of the Passerini moiety with the respective backbone moieties and side chains, and adding the result to respective

masses of the start and end fragments, the mass of the molecule can be calculated and used to identify the intact molecule in the MS spectrum. Similarly, the masses of the fragments can be calculated by the general formula provided in the supporting information (Chapter 3.5.1). Thus, the masses of the initial structure as well as the expected fragments can be calculated and traced in the MS/MS spectra, enabling the successful read-out of the sequence. It is important to note that both fragmentation pathways as well as the formulas provided in the supporting information are generally applicable to all herein discussed sequence-defined macromolecules, thus underlining the robustness and generality of the read-out.

Comments: MS/MS spectra are experimental data, not schemes. So, as previously recommended by one reviewer, MS/MS peaks used for sequence reconstruction should be labeled with m/z values. In their response to comments, the authors argue that adding m/z values in MS/MS spectra would make the figures confusing and, instead, list these values in separate tables. This is not a convenient presentation to the reader. Moreover, in the only MS/MS spectrum reported in the manuscript (Figure 5), there are only 11 useful peaks to be labeled with their respective m/z values, so adding these values will not complicate the figure that much (no need for 4 digit values, one decimal will be sufficient since accurate mass measurements are reported elsewhere).

Answer: Thank you for this comment. We have changed the diagrams in the manuscript and in the supporting information according to the reviewer's suggestions. Furthermore, we have removed the tables from the supporting information and have included the data in the spectra. We have added one spectrum as an example below (previously Figure 5 in the manuscript, now Figure 4):

Figure 1. Fragmentation of the dual sequence-defined pentamer **DS5** by tandem mass spectrometry, revealing the expected fragmentation pattern. A detailed peak assignment can be found in the supporting information. In the spectrum, the fragmentation next to the carbonyl group from both ends of the oligomer is depicted. By recombining the fragments, the initial structure of the pentamer can be re-established and thus the stored information is read.

Comments: Although most errors have been appropriately corrected, there are still a few issues with accurate mass data reported in SI (as listed below) and, although they are related to monomer synthesis, they must be fixed. In contrast, how can the authors explain that correcting calculated values had an influence on measured values? For example, changing the calculated m/z value of the 6d ion (p S20) from 180.1103 to 180.1019 has also changed the experimental 180.1104 to 180.1020!!!

Answer: Thank you very much. Indeed, there was a typing error in the original manuscript, which needed to be corrected during the previous revision and for which we apologise. We are happy to provide a picture of the original measurement below.

Comments: The same remark applies for M4 on p S22

Answer: Thank you. In this case, [M]⁺ was found instead of [M+H]⁺. HRMS was recalculated and we provide the original data below.

Comments: and for 6f on p S32.

Answer: Thank you. We corrected a typo and provide the original data below.

Comments: p S23: elemental composition of this ion is wrong: this should be C₁₁H₁₆NO₂⁺, not C₁₁H₁₅NO₂⁺

Answer: We have corrected that. Thank you!

Comments: p S25: the elemental composition is reported with no charge: do they author mean that this ion is not a protonated molecule but a molecular ion? If so, the charge should be “dot plus”.

Answer: Yes, thank you. We added the missing charge.

Comments: p S37: the reported experimental m/z value lacks precision.

Answer: Thank you. The high-resolution was missing here. We provide the original data and added the missing value.

Comments: p S39: calculated value for C33H56NO5 is 546.4153, not 546.4158.

Answer: Thank you. This is now corrected.

Comments: p S55: same remark as on p S25.

Answer: We added the missing charge.

REVIEWERS' COMMENTS:

Reviewer #1 (Remarks to the Author):

The authors have properly addressed all comments and issues raised by the reviewer: the revised version now reports a very complete study perfectly suited for the journal.